# Surface Roughness of Interior Fine Flow Channels in Selective Laser Melted Ti-6Al-4V Alloy Components

**DOI:** 10.3390/mi15030348

**Published:** 2024-02-29

**Authors:** Shamoon Al Islam, Liang Hao, Zunaira Javaid, Wei Xiong, Yan Li, Yasir Jamil, Qiaoyu Chen, Guangchao Han

**Affiliations:** 1Gemmological Institute, China University of Geosciences, Wuhan 430074, China; shamoonalislam@cug.edu.cn (S.A.I.); zunairajavaid@cug.edu.cn (Z.J.); xiongwei@cug.edu.cn (W.X.); yanli@cug.edu.cn (Y.L.); 20121001484@cug.edu.cn (Q.C.); hgc009@cug.edu.cn (G.H.); 2Advanced Manufacturing Research Institute, China University of Geosciences, Wuhan 430074, China; 3Hubei Jewelry Engineering Technology Research Centre, Wuhan 430074, China; 4School of Mechanical Engineering and Electronic Information, China University of Geosciences, Wuhan 430074, China; 5Laser Spectroscopy Laboratory, Department of Physics, University of Agriculture, Faisalabad 38040, Pakistan

**Keywords:** interior channel, additive manufacturing, overhang, space industry, process parameters optimization, laser powder bed fusion, build orientation, heat penetration, confined geometry, trapped powder

## Abstract

A challenge remains in achieving adequate surface roughness of SLM fabricated interior channels, which is crucial for fuel delivery in the space industry. This study investigated the surface roughness of interior fine flow channels (1 mm diameter) embedded in SLM fabricated TC4 alloy space components. A machine learning approach identified layer thickness as a significant factor affecting interior channel surface roughness, with an importance score of 1.184, followed by scan speed and laser power with scores of 0.758 and 0.512, respectively. The roughness resulted from thin layer thickness of 20 µm, predominantly formed through powder adherence, while from thicker layer of 50 µm, the roughness was mainly due to the stair step effect. Slow scan speeds increased melt pools solidification time at roof overhangs, causing molten metal to sag under gravity. Higher laser power increased melt pools temperature and led to dross formation at roof overhangs. Smaller hatch spaces increased roughness due to overlapping of melt tracks, while larger hatch spaces reduced surface roughness but led to decreased part density. The surface roughness was recorded at 34 µm for roof areas and 26.15 µm for floor areas. These findings contribute to potential adoption of TC4 alloy components in the space industry.

## 1. Introduction

Advancements in additive manufacturing (AM) have spurred transformative shifts across several engineering domains, especially in the space and high-performance sectors [1,2]. Over the years, substantial research efforts have been invested into enhancing the mechanical properties of AM metallic parts, ensuring they meet or even surpass the demands of traditional manufacturing methods. These advancements aim to produce lightweight internal cellular structures for structural components marked by strides in refining processes and material compositions, resulting in remarkable improvements in the structural integrity of AM components [3,4,5]. Despite these strides, a critical challenge, notably under-addressed in current literature, is the ability to construct complex internal geometries with precision, particularly in the context of interior fine fluid channels. The surface quality of these channels, which are indispensable in space and other high-performance applications, plays a vital role in the control of fuel delivery. In advanced space engineering, there is a growing movement towards the fabrication of embedded flow channels with reduced diameters within TC4 alloy space components. Specifically, in the millimetric scale, these channels are attributable to their superior thermo-fluid performance in space-restricted scenarios [6,7]. These dimensions facilitate efficient fluid dynamics, crucial for cutting-edge propulsion mechanisms.

The Selective Laser Melting (SLM) technique was utilized to fabricate internal lightweight cellular structures [3] and internal cooling channels in aerospace and combustion chamber components [8,9]. The proficiency of this technology was also extended to the creation of internal cooling channels for stainless steel injection molding tools, showcasing its versatility in diverse applications [10,11]. However, exploring SLM’s capabilities in fabricating embedded fine flow channels, with a focus on improving surface texture while maintaining the original CAD shape of TC4 alloy components, remains relatively uncharted. Ti-6Al-4V (TC4) titanium alloy, renowned for its exceptional strength-to-weight ratio, thermal stability, and corrosion resistance, is utilized in space and aerospace applications [12,13]. This alloy is notably employed in manufacturing complex components such as internal fluid channels, essential for creating multifunctional, high-performance spacecraft components [14]. The integration of these channels within titanium alloys parts contributes significantly to their functionality, enhancing performance efficiency in demanding aerospace environments [15,16]. The advanced AM techniques enable the precise formation of these intricate features, underscoring the importance of understanding TC4’s manufacturing intricacies for developing innovative aerospace designs with integrated fluid channels.

A few studies have been conducted to fabricate millimeter sized interior channels using this AM technique and each tried to control the channel shape deformation and improve interior surface texture [14]. They explored the effect of residual stress on channel shape deformation [17] but they did not consider the effect of the temperature field of trapped unhatched powder, which is more prominent in smaller sized channels. This is because powder that remains unhatched and confined within the tight regions is subjected to higher temperatures, a result of the heat emitted from recently build neighboring walls [18]. The metallic powder, in its heated and confined state, is more prone to attach to interior surface of the channel, resulting in higher roughness and poor texture [19]. This also leads to warping and increased roughness inside the SLM fabricated horizontal flow channels, especially on the underside of the overhang roof. The interaction of the laser with the powder bed during overhang construction is affected by keyhole fluctuations [20], leading to a deep, narrow cavity in the molten pool due to intense localized heating. This results in the deviations in the printed contour relative to the CAD model and increased roughness on the underside of the overhang roof [21,22]. Without a solid substrate to anchor it, the molten pool on an overhanging layer becomes unstable and its surface tension and fluid dynamics cause irregularities and a rougher finish compared to areas with a solid layer beneath [23,24,25]. Furthermore, channel roof overhangs lack this heat dissipation pathway, potentially causing inconsistent solidification and increased roughness [23,25,26]. The angle at which the laser strikes the overhang can also affects the melt pool’s behavior as non-optimal laser angles result in uneven melting and solidification, leading to a rougher surface texture [27,28]. Due to the lack of supporting material, the melt pool depth is more difficult to control on overhang regions. This causes the variable melt pool penetration and rough surface [29,30]. In conjunction with this, the newly formed layer, spanned across the powder without any underlying support, can lead to potential sagging or distortion [18,31]. As the heat dissipation becomes more difficult for the fine channel walls manufactured near the roof, the trapped unhatched powder within the confined regions of the channel experiences elevated temperatures due to the heat radiated by closely printed walls [18,31]. This hotter, confined powder is more prone to adhere to the channel roof as loosely bound material in contrast to powder on open-faced models, which can dissipate heat more effectively in free space. The internal curved faces of channels inherently display a stair-step effect that is not as pronounced on flat surfaces [31,32]. In conjunction with the steeper thermal gradients in internal channels due to confinement, the curved face becomes rougher due to the varied solidification rate compared to more openly accessible surfaces [33]. Melting pool dynamics in SLM are also varied between open face surfaces and confined geometries due to differences in heat dissipation, spatial confinement, and laser-material interactions [33,34]. Achieving uniform powder distribution is even more critical and challenging in small channels. Any inconsistency might lead to uneven melting and thus higher roughness [35,36].

Various studies have revealed that processing parameters play pivotal roles in optimizing surface roughness in SLM fabricated components. Response Surface Methodology (RSM) was employed to enhance the surface quality of Ti6Al4V parts fabricated through SLM, focusing on laser power, scanning speed, and hatch space [37]. Similarly, the influence of process parameters and their exposure times on roughness at upper and side surfaces of a Ti6Al4V specimen was also examined [38]. The upper surface’s roughness predominantly resulted from a ripple effect, while the inclined side surface was affected by both the step effect and powder adhesion. The investigations into different scanning strategies for Ti6Al4V specimens revealed that a helical scanning approach yields lower surface roughness compared to an S-shaped strategy [39]. It was also noted that surface roughness varied depending on the scanning direction. Scanning speed’s relationship with surface roughness was also explored, observing a parabolic trend where roughness initially increases and then decreases with increasing scanning speed, attributable to powder adhesion and droplet diffusion kinetics [40]. The effect of remelting scanning in a layer-by-layer approach on the roughness of the upper surface was investigated, which determined that selecting appropriate remelting parameters is crucial for reducing roughness [41]. The influence of specimen placement angle on surface roughness was also highlighted, noting differential roughness levels on upper and lower sides due to varying degrees of powder adhesion [42].

These studies underscored the significance of identifying optimal process parameters to enhance SLM part surface quality. However, few studies have been conducted to enhance the internal surface texture of TC4 alloy embedded flow channels in lightweight space components via SLM. For open surfaces, the post-printing machining processes, such as milling and burnishing, can be used to enhance the dimensional accuracy and surface quality of SLM parts [43,44]. The conformal cooling channels [45,46,47] represent a valuable application of SLM as they are critical in reducing cooling times and residual stresses in molded plastic parts. However, challenges arise in SLM due to the constraints in printing channels horizontally and without support, coupled with the inability to mechanically polish the interior surfaces due to structural limitations. It is difficult to fabricate horizontal cooling channels without supports and to predict the compensation for shape deviations in these channels [14,48]. The inside surface quality and roughness in overhanging areas are often the result of thermal stresses and damage due to overheating, exacerbated by inadequate thermal conductivity [49]. Studies have shown that channel orientation significantly impacts surface roughness in SLM printed interior channels, with vertically printed channels exhibiting the smoothest surfaces [50]. It is also indicated that the roof (top) roughness of horizontally printed channels often exceeds that of the floor (bottom) [16,51]. The top surfaces of channels typically exhibit higher roughness than the bottoms due to the inherent instabilities in overhanging layer deposition and the challenges in maintaining stable melt pools in these areas [22,50,52]. This distinction is critical, especially given the intricate nature of AM-produced channels and their impact on fluid dynamics and heat transfer efficiency. The higher surface roughness also influences parameters like friction factor and Nusselt number, with pronounced effects in channels with smaller hydraulic diameters [53]. The post-processing techniques, such as burnishing and abrasive flow machining (AFM), have been explored for their potential to refine these roughness disparities [43,54,55]. Notably, AFM shows promise in uniformly enhancing the surface quality, including both the top and bottom surfaces of internal channels [56]. Additionally, the process parameters, including laser power and scanning speed, play a crucial role in influencing the final part quality, affecting aspects like porosity and mechanical properties. Scanning strategies, divided into intra-layer and inter-layer strategies, further complicate the process. Horizontal accuracy is found to be unaffected by the angles of scanning tracks between layers [57]. However, the maximum hatch spacing is critical for achieving a smooth surface [58,59]. Despite these extensive studies, gaps remain in understanding and optimizing the surface roughness of small-diameter interior fluid channels, particularly with overhanging shape deviations.

The present study aims to investigate the influence of process parameters on the internal roughness and its correlation with the density of horizontally SLM 3D printed TC4 alloy fine flow fluid channels with a diameter of 1 mm. The most influencing factors, such as build orientation, layer thickness, scan speed, laser power, and hatch space, were investigated in order to simultaneously optimize those process parameters that yield low interior surface roughness and high part density.

## 2. Materials and Methods

### 2.1. Titanium Alloy Powder Materials

Gas-atomized, pre-alloyed Titanium alloy Ti6Al4V (TC4) spherical powder, with particle sizes ranging from 15–45 µm, was used in this research. The properties of the powder are detailed in Table 1. The powder was characterized using Bettersize 2000 under GB/T 19077 standard [60]. The powder was preheated to 100 °C for 1 h before SLM printing to ensure complete moisture removal [61], improve powder flowability [62,63], minimize thermal gradients [64], and stabilize the powder bed [65].

### 2.2. SLM Equipment and Preparation of Samples

Samples were fabricated utilizing the Selective Laser Melting (SLM) technique, employing a SISMA MYSINT100 system. This system features an Nd:YAG fiber laser with a 1070 nm wavelength, a maximum laser power capacity of 200 W, and a beam diameter measuring 30 μm. Error collection in SLM and important considerations taken during the experiment included were as follows: printing was performed perpendicular to the incident laser [66], a skywriting laser scanning strategy to ensure zero acceleration at boundaries and volume while hatching [67], and consideration of the printing orientation of parts with respect to recoater and gas flow direction [68,69]. All samples were built in an Argon environment (residual oxygen content was 0.5 vol.%), while the temperature and relative humidity (RH) of the build chamber were maintained at 25.6 °C and 33%, respectively. The orientation of the components, process parameters, and scanning strategy were developed using the AutoFab Mysint 2.0 by Materialise GmbH, Munich, Germany.

A single-track experiment was conducted using various laser powers and scan speeds to estimate the hatch space [70], as illustrated in Figure 1a; the corresponding process parameters are detailed in Table 2 (group a). The hatch space h was determined through Equation (1):(1)h=(1−Hr)×w
where w represents the width of the melt pool and Hr stands for melt pool overlap rate between the scanning track. Cubes with dimensions of 5 × 5 × 5 mm³ were produced using the process parameters from Table 2 (group b), depicted in Figure 1b. The relative density and surface roughness of the top surface of these cubes were measured [71]. Subsequently, a stack of cuboids, each measuring 6 mm × 6 mm × 4 mm, was printed. These cuboids were rotated at angles ranging from 0° to 90°, employing process parameters from Table 2 (group c) and shown in Figure 1c. Additionally, channel cuboids, measuring 5 mm × 5 mm × 10 mm (height, width, length) and featuring an internal channel with a 1 mm diameter, were fabricated using the parameters in Table 2 (group c). Each channel cuboid was printed at angles from 0° to 90° relative to the build direction, as shown in Figure 1d. Channel cuboids were printed horizontally with various layer thicknesses, laser powers, and scan speeds, detailed in Table 2 (groups d–g). To ensure uniform thermal gradient and homogeneous microstructure when optimizing process parameters, Meander scan strategy with a 90-degree angle of inclination at each layer was used in succeeding experiments [72,73,74,75]. Channel cuboids were again printed with optimized parameters and varying hatch spaces, detailed in Table 2 (group h).

### 2.3. Characterization Techniques and Analysis Tools

For characterization, a stylus-type profilometer (JB-6C, Shanghai Gaozhi Precision Instrument Co. Ltd., Shanghai, China) was used to measure the roughness (Ra) of the channel’s interior top (roof) and bottom (floor). Measurements were conducted three times at the center of both surfaces with a separation of 100 µm and a 4 mm stylus travel length. The average of these four Ra values was considered as roof and floor roughness. Further, the average of roof and floor roughness were again taken and considered as channel surface roughness. Microscopic images of the channels were captured and their roof and floor surface profiles were analyzed using a 3D digital microscopic (RX-100, HIROX, Co., Ltd., Tokyo, Japan). The volumetric density of the material was determined using the Archimedes technique. Percentage density was calculated by taking the ratio of the empirical density (obtained from measurements) to the theoretical density and expressing it as a percentage [76].

For analysis, thermal manufacturing simulations were conducted using Finite Element Analysis (FEA) with the commercial software Simufact Additive by Hexagon to study the temperature distribution and Surface deviation across the roof and floor of channels during the SLM building process. TC4 material properties critical to the simulation’s fidelity for SLM processes, including Young’s Modulus, thermal conductivity, specific heat, Poisson’s ratio, and density, were carefully determined through the rigorous property characterization tool, JMatPro V7.0 software. This involved aligning the material curves and constants for Ti6Al4V with those predefined by the manufacturer and meticulously determining them using JMatPro V7.0 software for properties including Young’s Modulus, thermal conductivity, specific heat, Poisson’s ratio, and density. The material was specified according to the Deutsches Institut für Normung (DIN) standard, with DIN number 3.7165 for Ti6Al4V. A comprehensive mesh sensitivity analysis was performed to determine the optimal mesh configuration that balances computational efficiency with simulation accuracy. Initially, the surface mesh was set at a relatively coarse element size of 1 mm, with volume mesh and voxel mesh settings adjusted correspondingly to offer a baseline for comparison. Successive iterations of the mesh refinement process involved decreasing the element size by 20% each step, examining the effects on key simulation outcomes such as temperature profiles and mechanical stress distributions. The analysis revealed a critical point of convergence where further mesh refinement resulted in less than a 3% change in simulated outcomes, indicating that an optimal mesh density had been achieved. This convergence point was reached with a surface mesh element size of 0.06 mm, a volume mesh element size of 0.5 mm, and a voxel mesh size of 0.05 mm in the x/y-plane. These configurations were chosen for their ability to accurately represent the geometry and thermal-mechanical behavior of the parts under study, with a total of 872 elements and 438 nodes for the surface mesh and 37,382 voxels for the voxel mesh, ensuring detailed and accurate modeling. Prior to simulations, default inherent strains calibration was employed, conducted with cantilever specimens. General key simulation parameters were hatch space 105 µm, laser power 120 W, scan speed 800 mm/s, layer thickness at 40 µm, chessboard scan strategy, and preheat temperature 100 °C. The default procedural approach spanned from geometry importation to the final post-processing steps, such as support removal and part separation from the build plate.

Principal Component Analysis (PCA) was applied to the data from Table 2 (groups d–g) to simplify the complex interactions among various SLM process parameters (laser power, scan speed, and layer thickness) into principal components to effectively capture the majority of the data’s variance [77,78]. This statistical tool is pivotal for identifying the parameters that significantly impact the channel surface roughness and material density of SLM fabricated channel interiors. Subsequently, the data from Table 2 (groups d–g) were further subjected to a robust Random Forest Machine Learning (RF-ML) regression model, employing a supervised learning approach. This model was trained on labeled data, where the target variable was the measured channel surface roughness associated with each set of process parameters (laser power, scan speed, and layer thickness) to closely examine the interactions between the parameters [79] and to optimize with increased accuracy [80,81]. The RF-ML model employed in this study was meticulously developed through a series of carefully structured steps, beginning with data preprocessing. Initially, the dataset was purified to ensure accuracy and consistency, with special attention paid to normalizing features to a common scale, thereby mitigating any potential bias due to variable scales. This preprocessing involved standardizing the laser power, scan speed, and layer thickness parameters to have a mean of zero and a standard deviation of one, ensuring that each feature contributed equally to the analysis. The Random Forest algorithm was configured with key parameters and hyperparameters to optimize performance. Specifically, the model utilized 150 trees to ensure a robust ensemble capable of capturing the complex relationships within the data without overfitting. The maximum depth of the trees was set to allow growth until all leaves were pure or contained less than five samples, ensuring detailed and nuanced decision-making capability within the model. The model employed bootstrapping to enhance the diversity among the trees and used the mean squared error criterion to measure the quality of splits during the tree-building process, prioritizing the reduction of variance within the data. Model training was conducted on a dataset divided into an 80% training set and a 20% testing set, ensuring both a substantial training dataset for model learning and an adequate testing dataset for model evaluation. Cross-validation, specifically a 5-fold cross-validation approach, was integral to the training process, allowing for the assessment and fine-tuning of the model’s hyperparameters, thus ensuring the model’s generalizability. The selection of the number of trees and the consideration for tree depth were critical components of this process, striking a balance between model complexity and computational efficiency. The model’s performance evaluation was based in a comprehensive suite of metrics, including R-squared (R²) for assessing the proportion of variance explained by the model, Mean Absolute Error (MAE), and Mean Squared Error (MSE) for quantifying the model’s prediction errors. These metrics facilitated a nuanced understanding of the model’s predictive accuracy and were instrumental in refining the model to achieve optimal performance. Post-processing of the model’s output involved a detailed analysis of feature importance, leveraging permutation importance techniques to identify and rank the significance of each process parameter in influencing the surface roughness. This analysis underscored the critical impact of layer thickness, laser power, and scan speed on the surface roughness outcomes, providing valuable insights into the process parameters optimization for improved SLM 3D printing results.

## 3. Results and Discussion

In single-track experiments, rectangular samples were printed as depicted in Figure 1a, utilizing the parameters specified in Table 2 (group a). The morphology of these samples was thoroughly examined and linked to the corresponding process parameters. The experiment involved sequentially scanning, layer-by-layer, a single track along the Z-axis to form each rectangular sample, resulting in the width of each sample being representative of a single weld pool [82]. These samples were observed via a digital microscope to estimate the thickness of each single track build under particular process parameters. A general increasing trend of higher melting was observed for scan tracks when the laser power was increased from 60 W to 120 W, while insufficient melting of powder particles was observed when scan speed was increased from 800 to 1800 mm/s at a fixed powder layer thickness of 40 µm. The most stable continuous single track of width 210 µm was recorded at laser power 120 W and scan speed 800 mm/s. The hatch space was determined to be 105 µm using Equation (1) with melt pool overlap rate Hr as 50%. Cubes measuring 5 × 5 × 5 mm³ were printed using this hatch space value of 105 µm at varying laser powers and scan speeds, using process parameters given in Table 2 (group b) and depicted in Figure 1b. The relative density and surface roughness (Ra) of the top surface of these cubes were measured. A laser power of 120 W, scan speed 800 mm/s, and layer thickness 40 µm yielded the lowest measured top surface roughness (Ra) of 18.5 µm and the highest relative density of 99%. This preliminary data served as the foundational process parameters for further experimentation aimed at optimizing the surface roughness of fine interior flow channels.

### 3.1. The Effects of Building Orientation on the Confined Surface Geometries

Cuboids printed in a stack, as described in Section 2.2 and Table 2 (group c), exhibited a significant variation in channel surface roughness relative to build orientation. As shown in Figure 2, the cuboid printed at a 0° orientation had a low surface roughness of 18.5 µm, which increased to 29.7 µm at a 10° build angle and further to 36.7 µm at 20°. The lower surface roughness at 0° is primarily due to a temperature gradient between the laser and solidifying material that balanced shear forces against surface tension, leading to smoother surfaces [83]. Additionally, the finer texture is aided by the overlapping curvature of individual melt tracks [84]. These combined thermal and geometric dynamics resulted in notably lower roughness at this angle. As the build angle increased beyond 0°, surface roughness also increases, largely because of the stair-step effect. The surface roughness pattern tended to increase between angles of 0° and 20°, then became linear at 30°. At 40°, surface roughness decreased to 28.1 µm and further to 22.8 µm at 50° due to the powder particles adhering to areas where loose powder meets the molten material, particularly at the outer edge of the contour scan track [83]. This powder adherence mechanism is assumed to smooth out the stair steps. Between 40° and 60°, surface roughness changed little, affected by both the stair-step effect and powder adherence mechanism [85], equating to the average size of the powder particles. Beyond 60°, surface roughness linearly dropped, reaching 13.5 µm at 90°. This is due to the decreasing stair-step effect with an increasing angle, with the lowest surface roughness at 90° occurring due to melt pool instability and thermal gradients [86,87].

When channel cuboids of Table 2 (group c) were printed with the same process parameters, the channel surface roughness showed an entirely different response. The floor roughness of horizontally printed channels was observed to be 76.8 µm, as shown in Figure 2. This increased floor roughness compared to the cuboid’s top surface at 0° is because the smaller diameter channel’s floor also exhibits a curved surface. Curved surfaces, particularly in smaller channels, are more affected by larger stair-step effects at 0° orientation which is parallel to build plate and powder adherence mechanisms, as illustrated in Figure 3a. For cuboids, heat is conducted through the entire powder bed during the manufacturing process, while in channel building, heat is confined, resulting in increased temperatures in trapped powder, which in turn increased powder adherence [18]. The roughness Ra at the roof of the channel was further increased to 92.2 µm, much higher than the size of powder particle.

The highest roughness (Ra) at the channel’s roof could be caused by the curved surface inducing larger stair-step effects, particularly larger overhang features when the roof is at 0° orientation, as depicted in Figure 3a. Furthermore, loose powder adhesion and the instability of the molten pool in overhanging layers, lacking solid substrate support, and efficient heat dissipation led to irregular solidification [23,25,26]. As the build angle was increased beyond 0°, the roughness (Ra) decreased abruptly to 67.3 µm at the roof and 54.8 µm at the floor of the channel at 30°. This reduction in both floor and roof roughness could be attributed to greater incline, which limited the stair step effect and overhang feature, while the overall decrease in the roughness (Ra) of both surfaces is possibly compensated by the attachment of loose powder, as shown in Figure 3b. The channel interior floor roughness at 30° remained higher than the open face of the cuboid’s plane surface because the curved interior cross-sections of the channel and hot trapped powder still contributed to channel surface roughness. Between 30° and 70°, roughness (Ra) changed little; at 70°, the roughness (Ra) at the roof was 61 µm and at the floor was 51.2 µm, showing the combined effects of stair-step and powder adherence mechanisms. However, at orientations beyond 70°, this value dropped abruptly to 38.5 µm at 90°, approximately equal at both roof and floor. This is because at these angles, both the stair-step and overhang wrapping effects are vanished, as shown in Figure 3c.

The observed higher channel surface roughness compared to the cuboid surface is due to the curved interior and the impact of hot, unhatched trapped powder. These findings underline the importance of heat management in SLM, particularly when dealing with complex internal geometries, such as fluid channels, which differ significantly from open, flat surfaces. These distinct geometric constraints and thermal behaviors within confined spaces emphasize the optimization of dedicated processing parameters. Such optimization is essential to achieve a uniform surface quality across varying geometries.

### 3.2. The Effects of Process Parameters on Channel Surface Roughness of the Horizontal Interior Flow Channels

Following the identification of the need for specialized process parameters in confined geometries, cuboids featuring a 1 mm diameter interior channel were printed with layer thicknesses of 20 µm, 30 µm, 40 µm, and 50 µm using varying laser powers and scan speeds, as detailed in Table 2 (groups d–g). The results revealed a complex interaction between the process parameters, channel surface roughness, and material density. Principal Component Analysis (PCA), a statistical tool, transformed this complex dataset into a smaller set of uncorrelated variables and identified patterns in the data by highlighting similarities and differences. The first three components accounted for over 95% of the total variation, indicating their significant influence, as shown in Figure 4a. Specifically, PC1 explained 53.6% of the variance, underscoring its primary effect, while PC2 and PC3 contributed 33.8% and 7.7%, respectively. This indicates that PC1 is responsible for separating the datasets, confirming discernible differences between the datasets. Figure 4b presents a 2-D plot showing the distribution of data according to PC1 and PC2, while Figure 4c presents a 3-D plot showing the distribution of the first three components. A layer thickness of 20 µm has a larger confidence ellipsoid with greater variability; less overlapping might be preferable for diverse outcomes and this thickness showed more distinct variations of parameters. The ellipsoids of layer thickness 30 µm, 40 µm, and 50 µm showed few variations, and more overlapping indicates that these groups are more consistent with each other. The distinct separation observed in the plot revealed clear differences in the results based on the process parameters.

Random Forest Machine Learning (RF-ML) modeling of the data from Table 2 (groups d–g) yielded the model’s performance, as indicated by the Root Mean Squared Error (RMSE) of approximately 9.09. This value reflects the average deviation of the predicted channel surface roughness values from the actual measurements in the dataset. The permutation importance graph, shown in Figure 5, revealed that layer thickness has a substantial effect on the channel surface roughness, with an importance score of 1.184. This suggests that variations in layer thickness significantly influence the surface quality of the printed channels, potentially due to the stair-step effect and variations in melt pool shape. In comparison, scan speed and laser power also showed notable importance, with scores of 0.758 and 0.512, respectively, indicating their roles in determining channel surface roughness, albeit to a lesser extent than layer thickness. The greater influence of scan speed over laser power could be due to the fact that scan speed directly controls the duration of laser interaction with the metal powder, thereby affecting the amount of energy deposited per unit time. This variation influences the consolidation rate, which is reflected in the channel surface roughness. Meanwhile, laser power governs the temperature gradient within the melt pool and the overall energy input, which in turn affects heat conduction into the adjacent powder and contributes to channel surface roughness due to powder adherence. These findings from the RF-ML analysis underscore the criticality of precise parameter control in SLM processes, particularly for achieving the desired surface quality in complex internal geometries like interior channels.

### 3.3. Effect of Layer Thickness on Channel Surface Roughness and Its Co-Relationship to the Density

The cuboids featuring interior channels of Table 2 (groups d–g) were analyzed for variations in channel surface roughness and density due to changes in layer thickness. Figure 6 illustrates the effect of layer thickness on the interior channel surface roughness (Ra) and part density at laser powers of 60 W and 120 W, with scan speeds ranging from 800 mm/s to 1800 mm/s. At a layer thickness of 20 µm, the channel surface roughness generally decreased with an increase in scan speed at the lower laser power of 60 W, as depicted in Figure 6a. Conversely, an increase in scan speed led to a decrease in part density, as seen in Figure 6b. The higher channel surface roughness at this smaller layer thickness was due to a greater number of laser hatches, which resulted in higher energy input into melting zone and its contained loose powders in the fine interior circular channels. The laser’s heat penetration through the thin layers and the heat accumulation within the roof area of the 1 mm circular channel combined to create a rougher surface with Ra of 63.97 µm for the slowest scan speed of 800 mm/s at a laser power of 60 W and laser energy density (LED) of 75 J/mm^3^, as shown in Figure 6a. Meanwhile, the thin layers experienced relatively longer consolidation times, leading to a higher part density of 97.01%, as illustrated in Figure 6b. When the laser power was increased to 120 W with same layer thickness and scan speed, the input LED raised to 150 J/mm^3^. This change led to a more pronounced heat penetration effect and increased the channel surface roughness to 69.61 µm and density to 99.6%, as shown in Figure 6c and Figure 6d, respectively. However, at higher scan speed of 1800 mm/s, the input LED was reduced to 33.33 J/mm^3^ due to a shorter consolidation time, resulting in less heat penetration into the adjacent powder. This limited the powder adherence, which resulted in a lower acceptable channel surface roughness of 33.72 µm but also reduced the part density to 93.99% at a laser power of 60 W. Similarly, a lower channel surface roughness of 44.4 µm and a decreased part density of 96.7% were observed with reduced LED 66.66 J/mm^3^ at a high laser power of 120 W with scan speed 1800 mm/s.

As layer thickness increased, the channel surface roughness significantly dropped to 43.6 µm at 30 µm, 37.30 µm at 40 µm, and 41.43 µm at 50 µm layer thickness for the samples fabricated at a laser power of 60 W and a scanning speed of 800 mm/s, as shown in Figure 6a. This general decrease in channel surface roughness of the 1 mm circular channel contrasts with the open cubic sample, where reduced layer thickness typically leads to decreased channel surface roughness due to the stair-step effect [88,89]. For the interior channels (the curved surface), thermal behavior related to solidification time, heat penetration, and containment within the confined area could have played a more influential role than the stair-step effect. Conversely, part density decreased with increasing layer thickness to 95.3% for a layer thickness of 30 µm, 91.8% for 40 µm, and 92% for 50 µm. This decrease was due to the reduction in LED to 41.66 J/mm^3^ for layer thickness 30 µm, 23.43 J/mm^3^ for 40 µm, and 16.66 J/mm^3^ for 50 µm, leading to insufficient melting of particles within the melting zone, as illustrated in Figure 6b. With a higher laser power of 120 W, a parabolic trend in channel surface roughness was noted. The channel surface roughness first decreased to 60.48 µm at 30 µm layer thickness, then to 46.09 µm at 40 µm, then abruptly increased to 81.59 µm at 50 µm, as shown in Figure 6c for samples prepared with a scanning speed of 800 mm/s. The lower channel surface roughness at the 30 µm and 40 µm layer thicknesses is presumed to result from adhered powder particles filling in the steps of the stair curves. At a layer thickness of 50 µm, which exceeds the largest powder particle size of 45 µm, the larger stair steps and overhangs at the roof are not adequately filled, leading to a higher deviation from the CAD contour and increased channel surface roughness, as depicted in Figure 7b. Despite this, part density slightly increased from 99.64% to 99.7% when the layer thickness was increased from 20 µm to 40 µm, but then dropped to 98.87% at 50 µm. This minor increase in density suggests that at smaller layer thicknesses, the higher laser power resulted in excessive energy density, causing spattering and pore formation, which subsequently reduced part density [90,91]. Increasing the layer thickness from 20 µm to 40 µm increased the depth of melt pool and decreased LED from 150 J/mm^3^ to 46.87 J/mm^3^, reducing spattering and pore formation, thereby enhancing density. However, at a layer thickness of 50 µm, the LED was too low at 33.33 J/mm^3^, leading to a decrease in part density. A similar parabolic trend in channel surface roughness was observed for samples prepared with a scan speed of 1000 mm/s. At higher scan speeds, this parabolic trend in channel surface roughness shifted, with a decrease in channel surface roughness at layer thickness 30 µm followed by an increase from layer thickness 40 µm and above. The channel surface roughness decreased to 35.42 µm at a layer thickness of 30 µm and then increased to 46.8 µm at 40 µm, followed by a rise to 71.2 µm at a 50 µm layer thickness for samples fabricated at a laser power of 120 W and scan speed 1800 mm/s. This initial reduction in channel surface roughness at a layer thickness of 30 µm can be attributed to the higher scan speed, which resulted in comparatively lower LED of 37.03 J/mm^3^, leading to reduced heat conduction in the confined powder of the channel, as reflected in the lower powder adherence and channel surface roughness. Conversely, at larger layer thicknesses of 40 µm and 50 µm with a slower scan speed of 1800 mm/s, insufficient melting of powders occurred due to too low LED 20.83 J/mm^3^ and 14.81 J/mm^3^ respectively, despite of higher laser power 120 W, leading to a rougher surface. This was evidenced by the lower part densities of 93.35% and 87.9%, respectively, as shown in Figure 6d. The results indicate that thermal behavior plays a larger role in channel surface roughness for relatively low layer thicknesses (20 µm to 30 µm) due to the adherence of loose powder, while the stair step effect has a greater influence on channel surface roughness for relatively high layer thicknesses (40 µm to 50 µm).

Figure 7c shows a 1 mm diameter interior channel horizontally printed at a layer thickness of 20 µm with a laser power of 60 W and a scan speed of 1000 mm/s, resulting in a smoother interior surface. In contrast, Figure 7d demonstrates the same channel printed with the same laser power and scan speed but at a layer thickness of 50 µm, displaying a poorer interior surface finish with more loose powder attached. The stair-step effect and powder adherence mechanism largely contribute to the channel printed at a layer thickness of 50 µm, resulting in a comparatively higher deviation of the printed contour from the CAD contour. Therefore, it was observed that smaller layer thicknesses of 20 µm not only ensure closer adherence to the CAD model due to smaller stair-step effects but also, if parameters are carefully optimized, they mitigate the challenges associated with heat management and stress accumulation, ultimately leading to higher density and superior surface quality within SLM-printed fluid channels.

### 3.4. Effect of Scan Speed on Channel Surface Roughness and Its Co-Relationship to the Density

The channel cuboids, printed horizontally using parameters from Table 2 (groups d–g) were examined to observe the impact of scan speed on channel surface roughness and material density at laser powers for layer thicknesses of 20 µm and 50 µm. At a scan speed of 800 mm/s and a layer thickness of 20 µm, channel surface roughness generally decreased with an increase in laser power, as shown in Figure 8a. However, it increased abruptly at very high laser powers. Conversely, as depicted in Figure 8b, an increase in laser power resulted in an increase in part density. At a scan speed of 800 mm/s, the channel surface roughness was 63.97 µm at laser power of 60 W, which decreased to 59.81 µm at 80 W and further to 43.6 µm at 100 W. The increasing laser power, along with the longer consolidation time at a slow scan speed, resulted in increased LED to 75 J/mm^3^, 100 J/mm^3^, and 125 J/mm^3^ respectively, which might have improved the melt pool’s stability and fluidity, favoring a uniform consolidation of the powder and resulting in a smoother surface. However, the heat penetration phenomenon in adjacent trapped powder still yielded powder adherence, resulting in the channel surface roughness being higher than that of open-face cubes, as discussed in Section 2.1. The part density showed an increasing trend at the slower scan speed of 800 mm/s due to the longer consolidation time and increasing laser power. The density increased from 97.01% to 99.64% with the laser power increasing from 60 W to 120 W; this was due to the gradual increase in input energy. At 120 W, the LED 150 J/mm^3^ was excessively high and the interaction of the higher laser power with the powder for a longer time at the slow scan speed led to splashing and a pronounced temperature gradient in the melt pool, which might have caused pore formation [90,92]. This, combined with the additional heat conducted into the loose trapped powder inside the 1 mm channel, might have collectively increased the channel surface roughness, as illustrated in Figure 9a. It was noted that as the number of scanning layers increased while building the channel’s floor and walls, heat penetration in the powder bed [18], especially localized inside trapped powder, promoted more powder adherence near the roof, resulting in an elevated channel surface roughness of 69.61 µm. A thermal simulation conducted using finite element analysis (FEA) also confirmed a pronounced temperature elevation within the channel, especially at the roof, as shown in Figure 9b. A temperature gradient was observed from the roof to the floor of the channel, suggesting different cooling rates and solidification patterns along the height of the channel. When the layer thickness increased to 50 µm, larger than the largest powder particle of 45 µm, the stair-step effect became more pronounced while the powder adherence mechanism was limited due to the decreased heat penetration in adjacent trapped powder from a smaller number of laser hatches along the Z-axis and decreased LED due to increase in depth of melting zone. This led to an opposite trend of channel surface roughness variation at the same scan speed for a layer thickness of 50 µm. The channel surface roughness under a scan speed of 800 mm/s increased from 41.43 µm to 81.59 µm as the laser power increased from 60 W to 120 W. At this greater thickness, the low laser power of 60 W and too low LED of 16.66 J/mm^3^ might not effectively penetrate through the entire layer, especially with fewer laser passes per layer, and might not conduct enough heat into adjacent trapped powder, resulting in channel surface roughness close to the size of the powder particles. The insufficient heat penetration could result in the presence of loosely bonded particles inside the curves of the stair steps. The insufficient melting of powder particles was also reflected in a decreased part density of 92.36%. As the laser power increased to 120 W under a slower scan speed of 800 mm/s and a larger inter-layer gap of 50 µm, the LED slightly increased to 33.33 J/mm^3^, which might have favored powder adherence, and the resultant elevated channel surface roughness could be due to the combined effect of the stair-step effect and powder adherence. The increased input energy also favored the complete melting of powder particles, resulting in an increased part density of 98.8%.

When the scan speed was increased, a general trend of decreasing channel surface roughness was observed for low laser powers of 60 W and high laser power of 120 W, while channel surface roughness fluctuated with moderate laser powers of 80 W and 100 W at a smaller layer thickness of 20 µm. This is because the increased scan speed limited the consolidation time, resulting in less heat penetration into adjacent powders. At an increased scan speed of 1000 mm/s with a low laser power of 60 W and a layer thickness of 20 µm, the melt pool consolidation time was reduced, lessening heat penetration and limiting powder adherence, which resulted in a decrease in channel surface roughness to 53.22 µm and part density to 96.50%, as shown in Figure 8a. At a higher scan speed of 1800 mm/s, the channel surface roughness decreased further to 33.72 µm due to the even shorter consolidation time that completely limited powder adherence. This shorter consolidation time also decreased LED to 33.33 J/mm^3^, resulting in incomplete melting of powder particles in the scanning layers and a reduced part density of 93.99%, as depicted in Figure 8b. At a higher laser power of 120 W with increased scan speed of 1000 mm/s, the shorter consolidation time decreased the LED to 120 J/mm^3^, leading to limited powder splashing and pore formation, reflected in a lower channel surface roughness of 65.79 µm compared to the channel surface roughness at a scan speed of 800 mm/s. This controlled powder splashing and reduced pore formation slightly increased the part density to 99.78%. At a higher scan speed of 1800 mm/s, the consolidation time was significantly reduced, limiting powder splashing and reducing the heat penetration phenomenon in the adjacent powder, resulting in less powder adherence and a reduced channel surface roughness (44.40 µm) with the same higher laser power. However, this shorter consolidation time reduced LED to 66.66 J/mm^3^, favoring incomplete melting of powder particles in the melt pool, which also decreased part density to 96.66%. For moderate laser powers, the effect of scan speed on channel surface roughness was less pronounced; as the scan speed increased from 1000 mm/s to 1800 mm/s, channel surface roughness fluctuated between 67.75 µm and 67.18 µm for a laser power of 80 W and between 58.50 µm and 54.19 µm for a laser power of 100 W. This is because the consolidation time reduced gradually as the scan speed increased, creating a balance between channel surface roughness due to irregular solidification from the increasing scan speed and limiting powder adherence due to the gradual decrease in heat penetration. However, part density gradually decreased with the decrease in input energy density, dropping from 97.81% to 93.56% for a laser power of 60 W and from 97.32% to 95.44% for a laser power of 100 W when the scan speed was increased from 1000 mm/s to 1800 mm/s with a layer thickness of 20 µm. At a higher layer thickness of 50 µm, increasing the scan speed from 1000 mm/s to 1800 mm/s showed a general increasing trend in channel surface roughness, as shown in Figure 8c. The larger inter-layer gap of 50 µm, coupled with the decreased input energy density from the increased scan speed, favored insufficient melting of powder particles and weaker bonding between layers. This irregular solidification promoted increased channel surface roughness for all given laser powers. Therefore, when the scan speed was increased from 1000 mm/s to 1800 mm/s, channel surface roughness increased from 40.31 µm to 47.40 µm for a laser power of 60 W, from 52.88 µm to 64.81 µm for a laser power of 80 W, from 56.02 µm to 72.33 µm for a laser power of 100 W, and from 64.86 µm to 71.20 µm for a laser power of 120 W. Similarly, the decreasing trend of insufficient melting also reduced part density from 81.90% to 66.39% for a laser power of 60 W, from 91.70% to 83.14% for a laser power of 80 W, from 97.56% to 93.04% for a laser power of 100 W, and from 98.64% to 87.95% for a laser power of 120 W, as shown in Figure 8d. The results indicate that consolidation time plays a significant role in the channel surface roughness of a 1 mm diameter channel by limiting powder splashing and the heat penetration phenomenon in the trapped powder of confined regions. Similarly, increasing scan speed generally appeared to decrease the part density as a result of decreasing input energy density.

Figure 9 presents two cross-sectional microscopic views of horizontally SLM 3D fabricated channels, illustrating the influence of scan speeds on the channel’s roof and floor surface profiles. At a lower scan speed of 800 mm/s with a laser power of 60 W and a layer thickness of 20 µm, the increased melt pool solidification time for the initial layers of the channel’s roof overhang allowed the molten metal more time to move and sag before complete solidification. This led to a droplet phase morphology, resulting in an uneven and rougher surface at the roof region, as depicted in Figure 9c. Conversely, at a higher scan speed of 1800 mm/s with the same laser power and layer thickness, the accelerated movement of the laser reduced the interaction time with the metal powder. This faster motion decreased the heat input, limiting excessive melting and deformation. Consequently, there was less sinking and sagging of the molten pool, resulting in a smoother and more uniform surface on the roof of the channel, as shown in Figure 9d. Therefore, it was observed that a higher scan speed is more effective for achieving a smoother surface finish on the roof overhang of the channel.

### 3.5. Effect of Laser Power on Channel Surface Roughness and Its Co-Relationship to the Density

The influence of laser power on the channel surface roughness and density of horizontally printed interior channel cuboids, with process parameters detailed in Table 2 (groups d–g), was investigated at layer thicknesses of 20 µm and 50 µm. At a laser power of 60 W and a layer thickness of 20 µm, the channel surface roughness generally decreased with an increase in scan speed, as shown in Figure 10a. Correspondingly, an increase in scan speed led to a decrease in part density, as seen in Figure 10b. Specifically, at a laser power of 60 W, the channel surface roughness was 63.97 µm at a scan speed of 800 mm/s. This channel surface roughness decreased to 53.22 µm at a scan speed of 1000 mm/s, to 54.24 µm at 1600 mm/s, and eventually to 33.72 µm at 1800 mm/s. When the scan speed increased from 800 mm/s to 1800 mm/s, the melt pool consolidation time decreased, reducing the LED from 75 J/mm^3^ to 33.33 J/mm^3^, which resulted in reduced temperature gradient of the melt pool and less heat penetration into adjacent powders. This process favored quicker solidification and resulted in less powder adherence, leading to a decrease in channel surface roughness. However, this limited temperature gradient caused insufficient melting of powder particles within the scanning areas, which was reflected in a decreased part density. Consequently, at a laser power of 60 W and a layer thickness of 20 µm, the part density decreased from 97.01% at a scan speed of 800 mm/s to 96.50% at 1000 mm/s, further to 94.40% at 1600 mm/s, and finally to 93.99% at 1800 mm/s. When the layer thickness was increased to 50 µm, the larger interlayer gap led to an increase in the depth of the melt pool, and the stair-step effect became more pronounced for a layer thickness that was larger than the size of the powder particles. Therefore, at a laser power of 60 W, the channel surface roughness variation with scan speed was not significant because LED was too low from 16.66 J/mm^3^ at scan speed 800 mm/s to 7.40 J/mm^3^ at 1800 mm/s. The channel surface roughness at a laser power of 60 W fluctuated between 41.43 µm and 47.30 µm for scan speeds of 800 mm/s to 1800 mm/s, as illustrated in Figure 10c. The lower input energy, a consequence of the higher layer thickness, was insufficient to fully melt the powder in the scanning areas, resulting in irregular melting. This was evidenced by a decrease in part density from 92.36% to 66.39% as scan speed increased from 800 mm/s to 1800 mm/s, as shown in Figure 10d.

When the laser power was increased to 80 W at a layer thickness of 20 µm, the higher LED of 100 J/mm^3^ with slower scan speed 800 mm/s resulted in an increased temperature gradient within the melt pool, which favored more complete melting of powder particles in the scanning layers, yielding a smoother surface. However, this also conducted more heat into the adjacent trapped powder, increasing powder adherence. At a laser power of 80 W, the observed channel surface roughness generally increased as the scan speed increased, as shown in Figure 10a. The channel surface roughness was recorded as 59.816 µm at a scan speed of 800 mm/s, 67.75 µm at 1000 mm/s, 67.78 µm at 1600 mm/s, and 67.182 µm at 1800 mm/s. The slower scan speed of 800 mm/s resulted in a longer consolidation time, which favored complete melting, smoother layer formation, and a higher part density of 97.42%, as shown in Figure 10b. The observed channel surface roughness at this scan speed was attributed to powder adherence. An increase in scan speed shortened the laser interaction time with the melt pool, leading to relatively quicker solidification. This rapid solidification might have increased channel surface roughness due to the quicker transition from the liquid to the solid phase and a higher temperature gradient between the melt pool and adjacent powders [93,94]. However, this quicker solidification of the melt pool also reduced porosity, resulting in a slight increase in part density to 97.81% at a scan speed of 1000 mm/s, 93.74% at 1600 mm/s, and 93.56% at 1800 mm/s, as cited in references [91,95]. When the layer thickness was increased to 50 µm with the increased laser power of 80 W, the channel surface roughness generally increased for higher scan speeds, as depicted in Figure 10c. A channel surface roughness of 52.88 µm was recorded at a scan speed of 1000 mm/s, 58.84 µm at 1600 mm/s, and 64.81 µm at 1800 mm/s. This increasing channel surface roughness was the result of pore formation and reduced LED favoring insufficient melting of powder particles in the melting zone due to the increasing consolidation at a greater interlayer gap of 50 µm. However, at the slower scan speed of 800 mm/s, the longer consolidation time resulted in an elevated channel surface roughness of 56.26 µm due to larger heat penetration into the trapped powder, which resulted in channel surface roughness due to powder adherence. This decreasing consolidation time with an increasing scan speed from 800 mm/s to 1800 mm/s resulted in decreasing part density from 93.30% to 83.14%, as shown in Figure 10d.

A similar trend of increasing channel surface roughness and decreasing part density with increasing scan speed due to rapid solidification of melt pools was observed at a laser power of 100 W and a layer thickness of 50 µm. The channel surface roughness increased from 53.03 µm to 72.33 µm when the scan speed was increased from 800 mm/s to 1800 mm/s, as shown in Figure 10c, for samples printed with a laser power of 100 W and a layer thickness of 50 µm. The part density decreased from 98.73% to 93.04% when the scan speed increased from 800 mm/s to 1800 mm/s due to decreasing LED from 27.77 J/mm^3^ to 12.34 J/mm^3^ favoring insufficient melting of powder particles in the melt pool, as seen in Figure 10d. However, at the smaller layer thickness of 20 µm, irregular solidification occurred at the higher laser power of 100 W due to the poor availability of powder particles within the smaller layer thickness (smaller than the average powder particle size of 30 µm). Therefore, the channel surface roughness variation in this case, with changing scan speed from 800 mm/s to 1800 mm/s, did not follow any particular trend and remained variable between 43.6 µm and 54.18 µm, as seen in Figure 10a. This increase in scan speed resulted in decreasing LED from 125 J/mm^3^ to 55.55 J/mm^3^ respectively, which promoted insufficient melting and eventually decreased part density from 98.15% to 95.44%, as seen in Figure 10b.

At a very high laser power of 120 W and a smaller layer thickness of 20 µm, both the channel surface roughness and part density decreased as the scan speed increased. The higher channel surface roughness at smaller scan speed of 800 mm/s was because the LED of 150 J/mm^3^ was too high, resulting in powder splashing due to a higher temperature gradient and a deeper keyhole effect, as shown in Figure 11a. This condition also caused more heat to penetrate in the trapped powders, increasing powder adherence. However, in the overhang areas of the channel roof, the higher laser power combined with a longer solidification time led to sinking and sagging, resulting in dross formation, as illustrated in Figure 11b. The channel surface roughness at this laser power and layer thickness was attributed to the combined effects of powder splashing, powder adherence, and dross formation at the roof. Consequently, channel surface roughness decreased from 69.61 µm to 44.40 µm when the scan speed was increased from 800 mm/s to 1800 mm/s, as shown in Figure 10a. The increasing scan speed reduced the laser interaction time with the metal powder, which favored quicker solidification and limited powder adherence, thus gradually reducing the channel surface roughness. On the other hand, the higher melting also improved bonding between scanning layers, resulting in an increased part density to 99.64% at a slow scan speed. Moreover, as the scan speed increased from 800 mm/s to 1800 mm/s, the LED decreased from 150 J/mm^3^ to 66.66 J/mm^3^, promoting insufficient melting and a decrease in part density from 99.64% to 96.66%. When the layer thickness was increased to 50 µm at a laser power of 120 W, the depth of the melt pool increased and the stair-step effect became more prominent. This condition limited powder splashing, but the heat may not have been able to fully penetrate and evenly distribute throughout the thicker layer, potentially leading to uneven melting and increased channel surface roughness [96]. Therefore, the channel surface roughness at a layer thickness of 50 µm was observed to be higher than that at a layer thickness of 20 µm because the deeper melt pool at the larger layer thickness also favored the entrapment of gases, contributing to increased porosity and lowering the density [94]. At a laser power of 120 W and a layer thickness of 50 µm, the channel surface roughness decreased when the scan speed increased due to a decrease in solidification time and lower heat penetration into the trapped powder, resulting in decreased powder adherence. As the scan speed increased from 800 mm/s to 1800 mm/s, the channel surface roughness decreased from 81.59 µm to 71.20 µm and the part density decreased from 98.87% to 87.95%, as shown in Figure 10c,d.

Figure 11c,d presents two microscopic images that illustrate the impact of laser power on the cross-sectional surface profiles of the roof and floor in horizontally SLM 3 D printed channels. When the laser power was increased from 60 W to 120 W, with a constant scan speed of 1800 mm/s and a layer thickness of 20 µm, distinct changes in the melt pool dynamics and the resulting surface morphology were observed. At a laser power of 60 W, the lower input energy combined with a higher scan speed of 1800 mm/s led to a melt pool that solidified very quickly. It interacted less with the trapped, unhatched powder within the channel, as shown in Figure 11c. The surface roughness at the roof was 35.4031 µm, which was still higher than that of the floor at 26.15 µm. This discrepancy is due to the molten pools at the roof overhang of the initial layers solidifying over unhatched powder layers, with fluctuating keyhole dynamics, as noted in reference [20]. This resulted in more sagging under gravity, as seen in Figure 11b. The hot trapped powder adhered more at the roof because it was at a higher temperature, having absorbed heat from the recently built channel’s floor and walls. Conversely, with a laser power of 120 W and the same scan speed of 1800 mm/s, as seen in Figure 11d, the higher input energy generated a larger and hotter melt pool. This led to sinking and sagging under gravity, resulting in dross formation at the roof, and conducted more heat into the trapped powder, increasing the tendency for loose powder adherence. The surface roughness at the roof, recorded at 65.16 µm, was much higher than that at the floor, which was 23.03 µm with this laser power of 120 W. During the SLM building process, the higher input energy ensured full melting of powder particles and efficient fusion of the scanning layer with the preceding layer, resulting in a smoother surface at the floor. However, when building the roof overhang layers, the hotter and denser molten pools sagged under gravity, forming dross and warping under residual stress accumulation, as further exacerbated by the adherence of hot, trapped, unhatched powder [97]. It was observed that a laser power of 60 W combined with scan speed 1800 mm/s and layer thickness of 20 µm yielded reduced surface roughness at roof and floor, and maintaining part density in SLM 3D printed horizontal channels. This specific parameter set minimizes powder adherence and promotes rapid yet controlled solidification, leading to smoother channel surfaces and consistent part quality.

### 3.6. Effect of Hatch Space on Channel Surface Roughness and Its Co-Relationship to the Density

The process parameters from Table 2 (groups d–g) simultaneously yielded relatively lower channel surface roughness, a finer surface finish, and high part density; specifically, a layer thickness of 20 µm, laser power of 60 W, and scan speed of 1800 mm/s were further investigated with varying hatch spaces as detailed in Table 2 (group h). As depicted in Figure 12a, the surface roughness at the roof of the horizontally printed 1 mm diameter channel generally decreased as hatch space increased, while the floor roughness was much higher at smaller hatch spaces. The part density also decreased with an increase in hatch space, as seen in Figure 12b. At a smaller hatch space, the greater overlapping of adjacent melt pools led to excessive melting and irregular solidifications, as can be seen in the channel floor in Figure 13a. Additionally, at smaller hatch spaces, the input energy density was 55.55 J/mm^3^, leading to more heat penetration into adjacent unhatched trapped powders, resulting in higher powder adherence and a poorer surface finish at the floor. In addition to irregular solidification and heat-induced powder adherence, the hotter overlapped melt pools at roof overhang regions sagged under gravity, as seen at the roof in Figure 13a. The surface roughness at the smaller hatch space of 30 µm was 44.81 µm at the roof and 41.82 µm at the floor. This higher overlap of melt pools resulted in excessive melting of powders, achieving a higher part density of 98.85%.

When the hatch space was increased to 40 µm, it reduced the overlap of adjacent melt pools, which in turn reduced the input energy density to 41.66 J/mm^3^, limited powder adherence due to less heat penetration into adjacent trapped powders, and resulted in a smoother surface at the floor of the channel, as shown in Figure 13b, with Ra of 26.15 µm. When the roof was built, the melt pools solidified before sagging due to decreased input energy with higher scan speed, and the limited powder adherence collectively resulted in comparatively less surface roughness at the roof than the roof built with a smaller hatch space, as seen in Figure 13b. Therefore, at a hatch space of 40 µm, the surface roughness was observed as 26.15 µm at the floor and 34.03 µm at the roof of the channel, as seen in Figure 12a. However, due to the decreased energy density, the part density was also reduced to 98.62%, but it remained close to the density at a hatch space of 30 µm. This indicates that a hatch space slightly greater than the laser’s focus diameter of 30 µm and the average powder particle size of 30 µm could facilitated optimal overlap between adjacent melt pools. This optimal overlap ensured that the gaps between the scanned lines are filled effectively, reducing porosity and maintain the part density.

When the hatch space was further increased to 50 µm, this larger hatch space could result in less overlap between adjacent scan tracks, potentially leading to areas of partially melted powder. Additionally, a larger hatch space resulted in less thermal overlap, which might affect melt pool dynamics, leading to inconsistent melting and solidification that increased roof roughness to 35.01 µm and floor roughness to 32.43 µm, as shown in Figure 12a, with the respective micrographs depicted in Figure 13c. Conversely, as the input energy density was reduced to 33.33 J/mm^3^, which provided less consolidation of powder particles, leading to a structure with more voids and a decreased part density of 93.99%. When the hatch space was increased to 60 µm, it was too large to provide adequate melting and fusion of the particles in melt pools, potentially compromising mechanical integrity despite a smoother appearance, as seen in Figure 13d. The surface roughness at the roof and floor due to insufficient melting was observed as 31.04 µm and 29.10 µm, respectively. It can be noted that larger hatch spaces improved surface quality by reducing the amount of loose powder that fails to fuse. However, the input energy density decreased to 27.77 J/mm^3^, which in turn reduced the part density to 88.64%, as shown in Figure 12b. A decline in density with increasing hatch space indicates that larger spaces result in less overlap between melt tracks, leading to a structure with more voids. This decrease in density correlated with the observed improvement in surface smoothness, implied less powder adherence, which generally resulted in a smooth surface. Since the powder particle size ranges from 15 µm to 45 µm, a hatch space that is too large could prevent adequate melting and fusion of the particles, potentially compromising mechanical integrity despite a smoother appearance. However, the results indicated that a hatch space of 40 µm, in combination with the parameters specified in Table 2 (group h), simultaneously achieved acceptable roof and floor roughness along with a higher part density.

## 4. Conclusions

This study is the first of its kind on fabricating interior small diameter (1 mm) circular fluid channels horizontally through Selective Laser Melting (SLM) and investigating the effects of process parameters on the internal surface roughness including the roof and bottom areas. It should provide valuable insights into how to deliver low surface roughness and higher density and optimize process parameters for specialized TC4 alloy interior confined structural components. The key findings are:Build orientation was identified as a crucial factor influencing surface roughness in SLM-printed open-faced cuboids and interior 1 mm circular channels. At 0°, cuboid surface roughness resulted from the overlapping of curvatures of individual melt tracks and temperature gradients, which increased at steeper angles due to stair step effects and reduced to the lowest at 90°. Conversely, channel surface roughness was highest at 0° (horizontal direction), exacerbated by the sagging of melt tracks at roof overhangs, and by stair step effects and heat-induced powder adherence at the roof and floor. It decreased at steeper angles due to reduced overhangs and stair step effects, yet heat-induced powder adherence remained constant. It decreased to the lowest at 90°, at both roof and floor (exhibiting equal surface roughness).A machine learning based analysis using Principal Component Analysis for dimensionality reduction, followed by Random Forest Regression, identified significant impacts of process parameters on channel surface roughness. Layer thickness was indicated a critical parameter, with higher importance score of 1.184, while scan speed and laser power also had notable impacts, scoring 0.758 and 0.512, respectively.Channel surface roughness was markedly affected by layer thickness. Ra at small layer thicknesses (20 µm), smaller than the average powder particle size (30 µm), was primarily influenced by powder adherence, exacerbated by increased number of laser hatches along z-axis and input higher energy. Conversely, at thicker layer thicknesses (50 µm), channel surface roughness was mainly caused by a combination of stair-step effects and powder adherence. It was observed that a smaller layer thickness of 20 µm yielded a finer interior surface than a larger layer thickness of 50 µm.The quality of the horizontal channel’s roof was significantly affected by scan speed. At a smaller layer thickness of 20 µm, slow scan speeds increased melt pool consolidation time at roof overhangs, causing the molten metal to sag before fully solidifying, leading to uneven and rougher surfaces but resulted higher part density. Faster scan speeds reduced the laser interaction time with the metal powder, decreasing heat input and limiting excessive melting and deformation, resulting in smoother surfaces but relatively lower part density. At a larger layer thickness of 50 µm, with reduced input energy due to a larger interlayer gap and pronounced stair-step effect, changing scan speed resulted in relatively higher channel surface roughness. However, part density decreased sharply as scan speed increased compared to smaller layer thickness.Laser power also contributed to internal channel surface roughness by increasing the melt pool temperature and accumulating heat inside the trapped powders of the channel. At a smaller layer thickness of 20 µm and low laser power, an increase in scan speed resulted in a decrease in channel surface roughness and part density by reducing laser interaction time with metal powder. At higher laser power with smaller layer thickness, the excessive input energy caused powder splashing and irregular melting, and led to sinking and sagging of melt pools at roof overhangs of horizontal channels, resulting in warping and dross formation. At a larger layer thickness of 50 µm, the increased interlayer gap limited powder splashing at higher laser power but decreased input energy. Generally, an increase in laser power resulted in an increase in both channel surface roughness and part density due to the combined effect of heat-induced powder adherence and the stair-step effect at larger layer thickness.Varying hatch space influenced the channel surface roughness and part density. Smaller hatch spaces increased channel surface roughness due to greater overlap of melt tracks and higher input energy, resulting in warping at the roof and irregular solidification at the floor of the horizontal channel. Conversely, very large hatch spaces reduced channel surface roughness through less heat penetration but also decreased part density. It was observed that at a layer thickness of 20 µm, laser power of 60 W, and scan speed of 1800 mm/s, a hatch space slightly greater than the laser’s focus diameter of 30 µm, and the average powder particle size of 30 µm could facilitate optimal overlap between adjacent melt pools. This optimal overlap ensured that gaps between scanned lines were effectively filled, reducing average channel surface roughness and maintaining part density.

## Figures and Tables

**Figure 1 micromachines-15-00348-f001:**
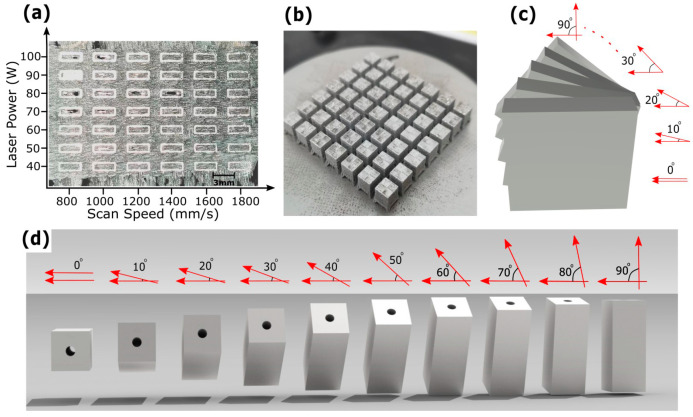
SLM Sample Preparation and Experimental Design: (**a**) single scan tracks for hatch space estimation; (**b**) cubes produced for density and surface roughness measurements; (**c**) cuboids oriented at varying angles for surface roughness assessment; (**d**) channel cuboids designed for orientation dependent printing.

**Figure 2 micromachines-15-00348-f002:**
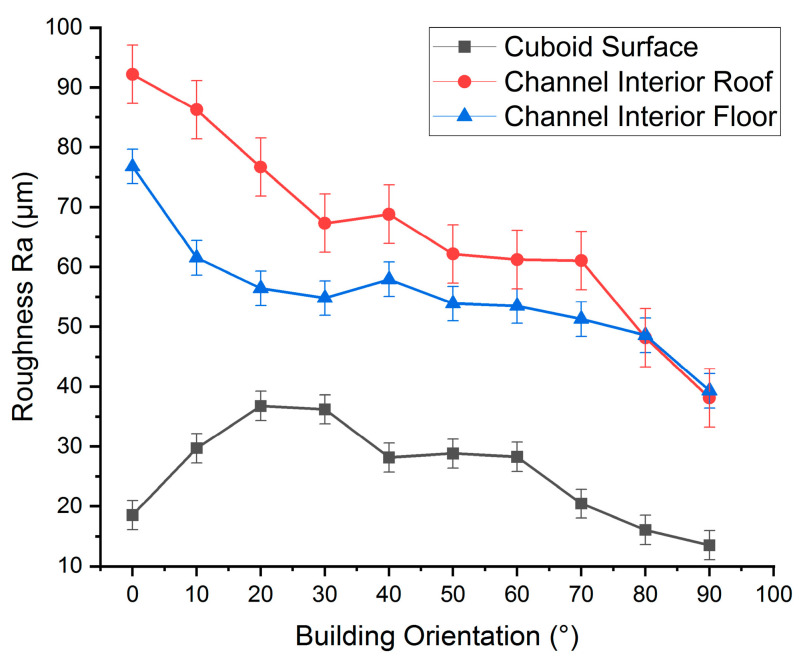
Building orientation effect on surface roughness of cuboid open surfaces and channel interior roof and floor.

**Figure 3 micromachines-15-00348-f003:**
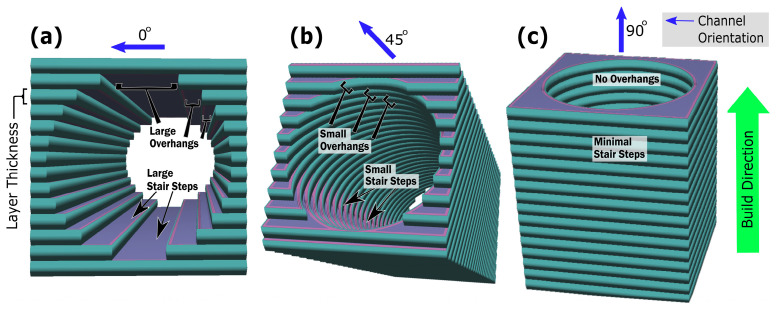
An illustration of impact of build orientation on channel surface roughness: (**a**) layer slicing horizontal channels at 0° orientation resulting larger overhang areas and stair steps; (**b**) channels at 45° orientation with smaller overhang areas and smaller stair steps; (**c**) vertical channels at 90° orientation exhibits no overhangs and minimal stair steps.

**Figure 4 micromachines-15-00348-f004:**
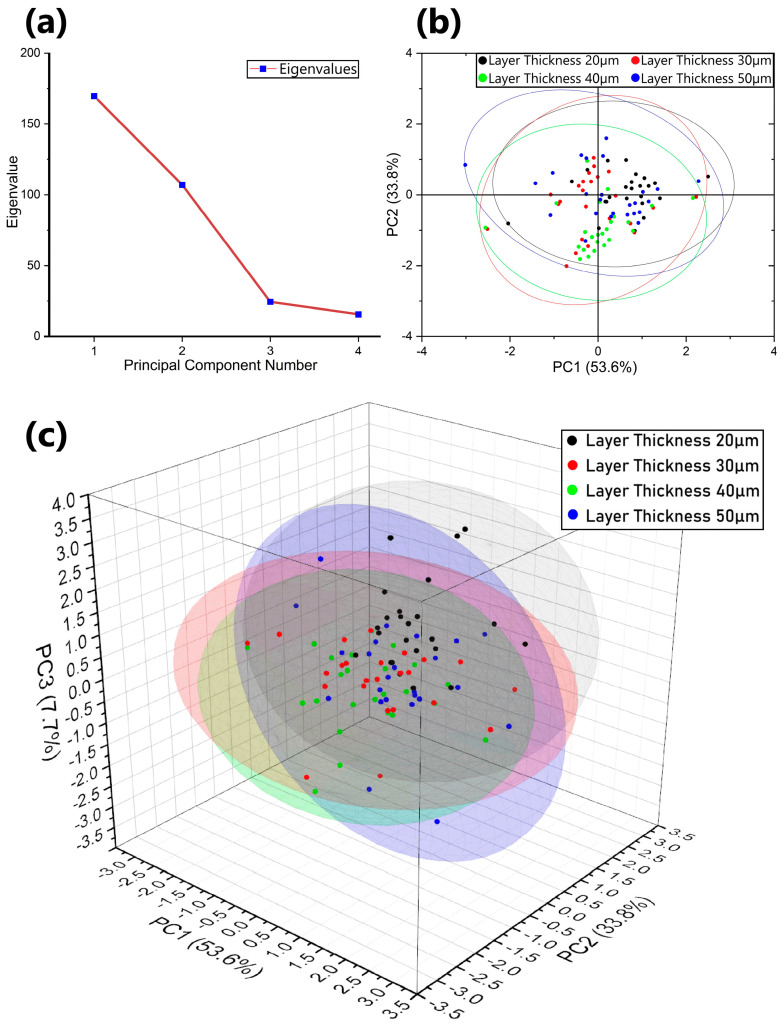
Principal Component Statistical Analysis: (**a**) loading plot; (**b**) 2-D score plot of the first two principal components containing a cumulative 87.4% variance; (**c**) 3-D score plot of the first three principal components containing a cumulative 95% variance.

**Figure 5 micromachines-15-00348-f005:**
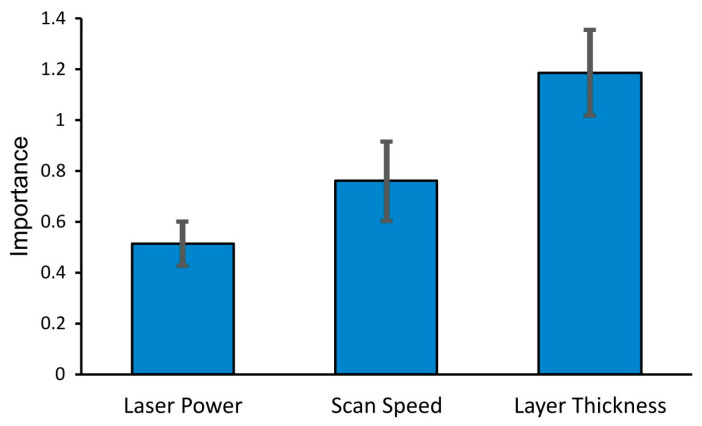
Permutation Importance of Process Parameters on Channel Surface Roughness.

**Figure 6 micromachines-15-00348-f006:**
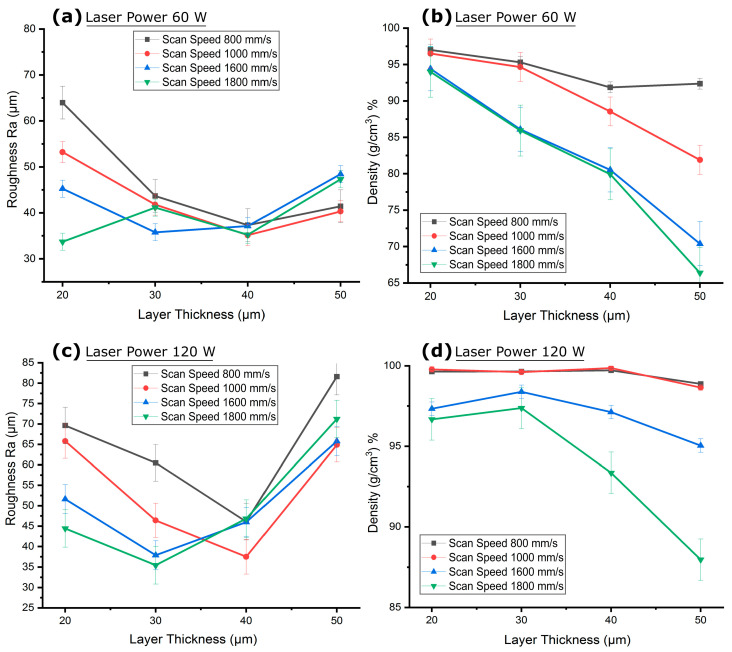
Effects of layer thickness on the channel surface roughness and density of 1 mm fluidic channels at different scan speeds: (**a**) channel surface roughness variation at 60 W laser power; (**b**) density variation at 60 W laser power; (**c**) channel surface roughness variation at 120 W laser power; (**d**) density variation at 120 W laser power.

**Figure 7 micromachines-15-00348-f007:**
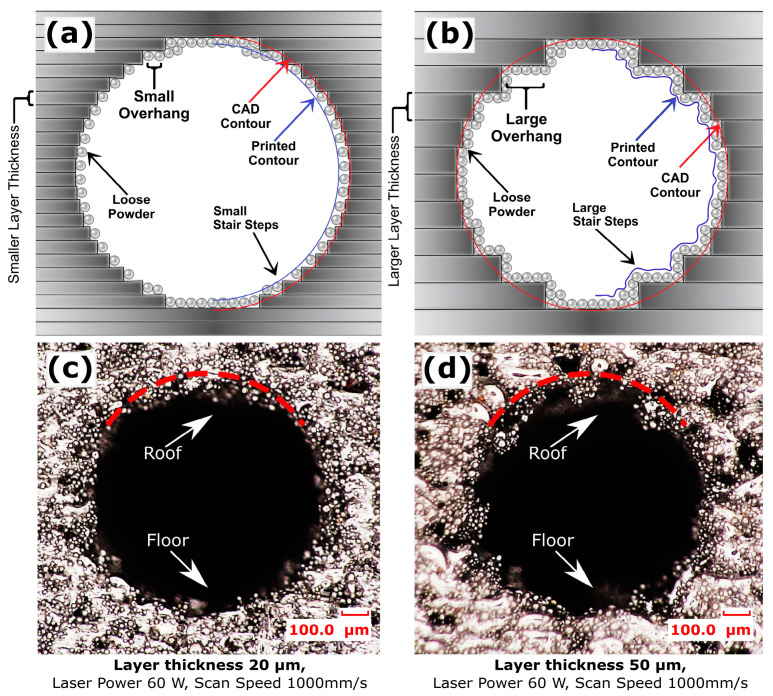
Impact of layer thickness on surface finish and overhangs: (**a**) small layer thickness leading to more pronounced channel surface roughness due to powder adherence, yielding a finer surface; (**b**) Large layer thickness resulting in increased channel surface roughness due to large stair steps and more significant overhang areas; (**c**) Optical microscopy image of a channel printed with a 20 µm layer thickness, showing a relatively finer inside surface; (**d**) Optical microscopy image of a channel printed with a 50 µm layer thickness, displaying poorer surface characteristics.

**Figure 8 micromachines-15-00348-f008:**
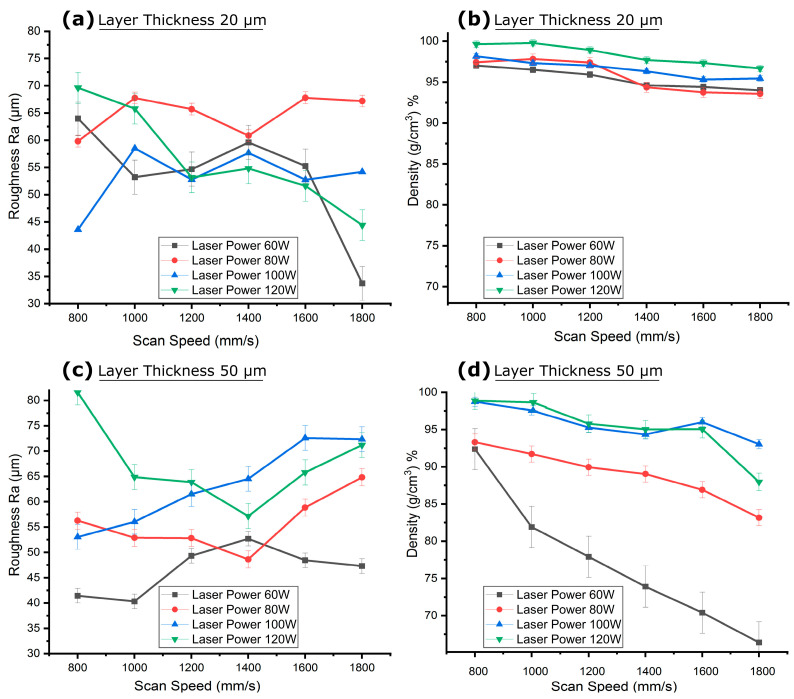
Effects of scan speed on the channel surface roughness and density of interior channels: (**a**) channel surface roughness variation at various scan speeds for 20 µm layer thickness; (**b**) density variation at various scan speeds for 20 µm layer thickness; (**c**) channel surface roughness variation at various scan speeds for 50 µm layer thickness; (**d**) density variation at various scan speeds for 50 µm layer thickness.

**Figure 9 micromachines-15-00348-f009:**
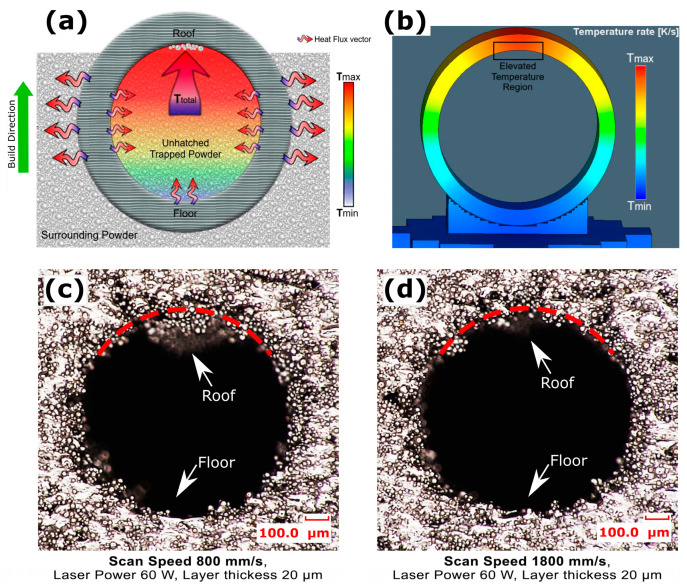
Factors affecting 1 mm diameter horizontally printed channel surface quality and overhang roof: (**a**) heat penetration into trapped powder increases powder adherence, especially at the channel roof; (**b**) FEA simulation of the channel’s manufacturing process showing a thermal gradient along the z-axis with elevated temperatures at the roof; (**c**) Optical microscopy image of a horizontal interior channel’s cross-sectional surface profile at scan speed 800 mm/s, laser power 60 W and layer thickness 20 µm, illustrating droplet phase formation at the roof due to sagging from longer consolidation times; (**d**) Optical microscopy image of a horizontal channel surface profile at a higher scan speed 1800 mm/s with same laser power and layer thickness, showing reduced roof sagging due to shorter consolidation time.

**Figure 10 micromachines-15-00348-f010:**
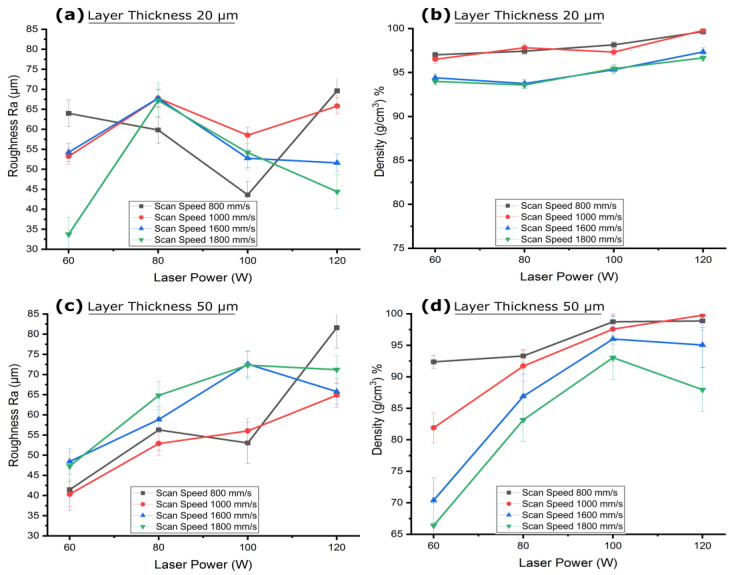
Effects of laser power on the channel surface roughness and density of interior channels: (**a**) Channel surface roughness variation at various laser powers for layer thickness of 20 µm; (**b**) Density variation at various laser powers for layer thickness of 20 µm; (**c**) Channel surface roughness variation at various laser powers for layer thickness of 50 µm; (**d**) Density variation at various laser powers for a layer thickness of 50 µm.

**Figure 11 micromachines-15-00348-f011:**
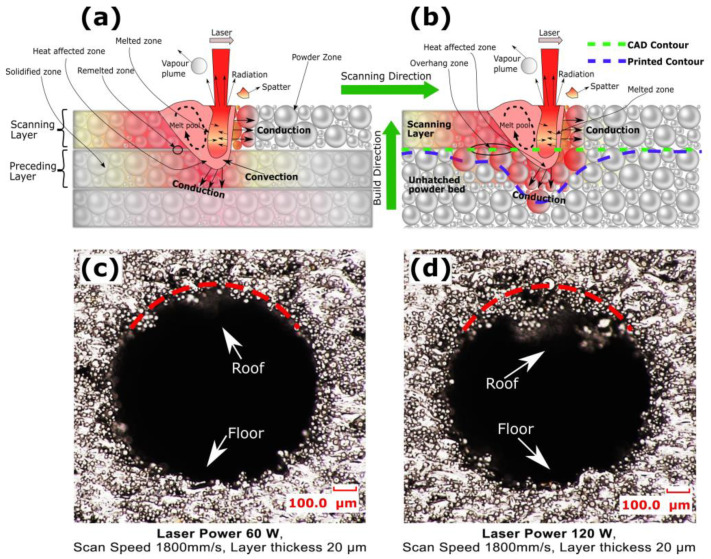
Factors affecting floor and roof formation in SLM printed horizontal channels: (**a**) laser scanning and welding process of scanning layers onto the preceding floor layer; (**b**) Overhang deviation during roof formation over the unhatched powder zone; (**c**) Optical microscopy image of the cross-sectional channel surface profile at a laser power of 60 W, scan speed of 1800 mm/s, and layer thickness of 20 µm; (**d**) Optical microscopy image at a higher laser power of 120 W printed with the same parameters.

**Figure 12 micromachines-15-00348-f012:**
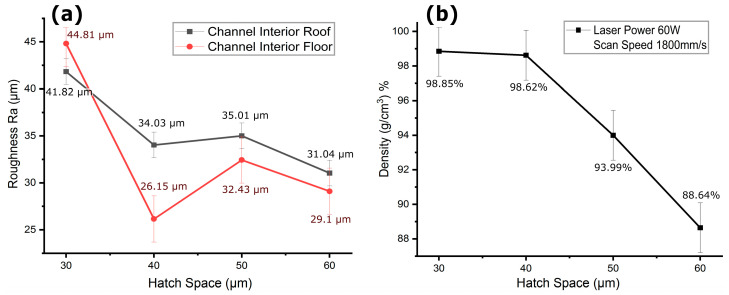
Effect of hatch space on surface roughness and density of SLM 3D horizontally printed 1 mm diameter interior channel: (**a**) roof and floor roughness variation at various hatch spaces, for sample printed with laser power 60 W, scan speed 1800 mm/s and layer thickness 20 µm; (**b**) density variation at various hatch spaces.

**Figure 13 micromachines-15-00348-f013:**
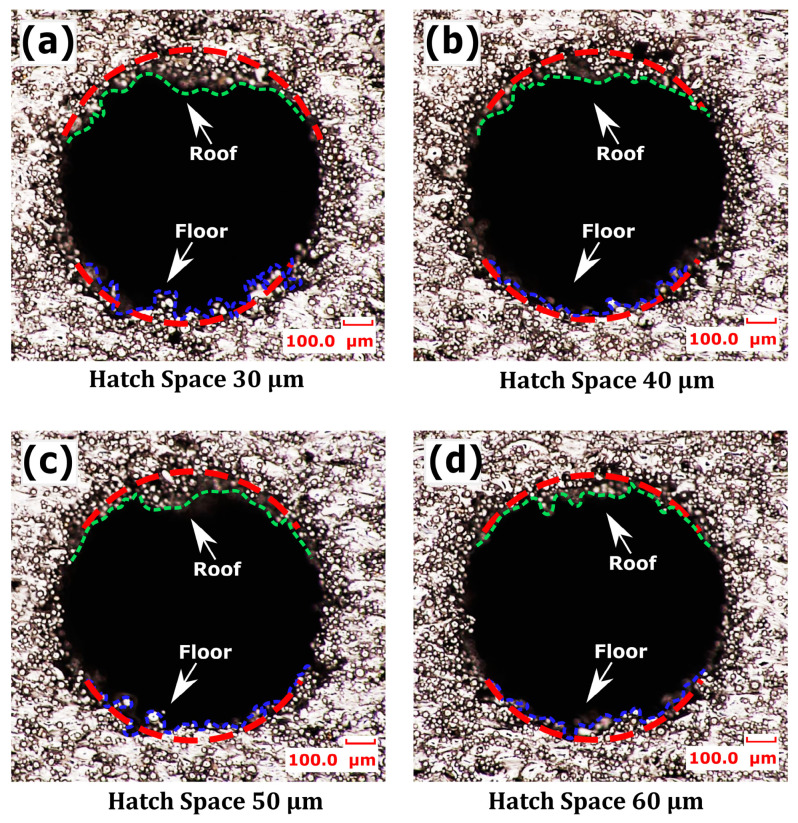
Optical microscopy of horizontal interior channel’s cross-sectional surface, showing roof and floor profiles at laser power 60 W, scan speed 1800 mm/s and layer thickness 20 µm across hatch spaces from 30 µm to 60 µm (**a**–**d**). The red colored dotted lines represent the CAD-designed contours within the channel, the green dotted lines illustrate the upper boundary of the printed contour at the roof and the blue dotted lines show the lower boundary of the printed contour of the floor. This delineation highlights the variations in the surface profile attributable to changes in hatch space.

**Table 1 micromachines-15-00348-t001:** Powder properties.

Powder Composition (%wt.)									Size Distribution (μm)		
Al	V	Ti	Fe	Si	O	C	N	H	d10	d50	d90
5.89	4.11	balance	<0.04	<0.02	<0.0706	<0.012	<0.011	<0.0006	18.48	28.79	45.68

**Table 2 micromachines-15-00348-t002:** Process parameters for SLM experiments, including single tracks, cubic blocks, and channels printed at various angles and layer thicknesses.

Group	Model Geometry	Layer Thickness t (μm)	Laser Power P (W)	Scan Speed S (mm/s)	Hatch Space h (μm)	Scan Strategy	Build Orientation
a	Singe Tracks	40	60–120	800–1800	-	-	-
b	Cubes	40	60–120	800–1800	105	Chessboard	-
c	Cuboid stack and Channel Cuboids	40	120	800	105	Chessboard	0°, 10°, 20°,…,90°
d	Channel Cuboids	20	60–120	800–1800	50	Meander	0°
e	Channel Cuboids	30	60–120	800–1800	60	Meander	0°
f	Channel Cuboids	40	60–120	800–1800	80	Meander	0°
g	Channel Cuboids	50	60–120	800–1800	90	Meander	0°
h	Channel Cuboids	20	60	1800	30, 40, 50, 60	Meander	0°

## Data Availability

Data are contained within the article.

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
