# Peer review of "Surface Roughness of Interior Fine Flow Channels in Selective Laser Melted Ti-6Al-4V Alloy Components"

_micromachines, 2024, doi:10.3390/mi15030348_

Round 1
Reviewer 1 Report
Comments and Suggestions for Authors
In this work, the authors studied Surface Roughness of Interior Fine Flow Channels in Selected Laser Melted Ti-6Al-4V Alloy Components. The paper is interesting and well organized. The corresponding literature analysis is also quite comprehensive. However, there are still some questions needing to be solved before accepted.
1. The mechanism of influence of laser energy density on surface quality of Ti-6Al-4V alloy can be further discussed. The following reference can be referred.
(Experimental study on mechanism of influence of laser energy density on surface quality of Ti-6Al-4V alloy in selective laser melting [J]. Journal of Central South University, 2022, 29(10): 3447-3462. DOI: https://doi.org/10.1007/s11771-022-5135-1)
2. The author mentions that the surface roughness was tested 3 times, so an error bar should be included in Figure 2.
3. The 3D arrangement of Figure 4c is difficult for the reader to discern, and the horizontal coordinate is obscured from view.
4. The horizontal coordinates of Figure 8,9,10, and 12 are not shown.
Author Response
Title: Surface Roughness of Interior Fine Flow Channels in Selective Laser Melted Ti-6Al-4V Alloy Components
Shamoon Al Islam1,2,3, Liang Hao1,2,3,*, Zunaira Javaid1,2,3, Wei Xiong1,4, Yan Li1,2,3, Yasir Jamil5, Qiaoyu Chen1,2,3 and Guangchao Han1,2,4
Response: We deeply appreciate the time and effort you have invested in reviewing our manuscript. Your detailed and insightful feedback has significantly contributed to the enhancement of our work. Following your valuable suggestions, we have made the recommended changes. We hope that the revisions now accurately reflect the improvement.
Comment -1: The mechanism of influence of laser energy density on surface quality of Ti-6Al-4V alloy can be further discussed. The following reference can be referred.
(Experimental study on mechanism of influence of laser energy density on surface quality ofTi-6Al-4V alloy in selective laser melting []. journal of Central South University, 2022, 29(10)3447-3462.D0l:https:/doi.org110.10071s11771-022-5135-1)
Response -1: Thank you for highlighting the influence of laser energy density (LED) on the surface quality of SLM-manufactured interior channels. In response, we have made corresponding enhancements to the discussion sections. For instance, Section 3.3 now elaborates on the effects of layer thickness on surface roughness and density. We explained that at a smaller layer thickness of 20 µm, with low laser power of 60 W and a slow scan speed of 800 mm/s, the reduced melt pool volume resulted in a LED of 75 J/mm3. This facilitated more complete melting of powder particles within the melting zone, leading to a higher part density of 97.01% and a surface roughness of 63.97 µm. Conversely, at a larger layer thickness of 50 µm, under the same laser power and scan speed, the increased melt pool volume led to a decreased LED of 16.66 J/mm3, resulted in incomplete melting of powder particles and a reduced part density of 92.36%. However, the lower LED also causes less heat penetration into the trapped powder of channel’s confined region, reducing powder adherence and thus decreasing the channel surface roughness to 41.43 µm. This detailed analysis clarifies the impact of LED variations due to layer thickness adjustments on the channel’s surface roughness, indicating that a lower layer thickness can promote a smoother finish due to better energy absorption and material consolidation. In Section 3.4, we further discuss how at a higher laser power of 120 W and scan speed 800 mm/s and layer thickness 20 µm, an excessively high LED of 150 J/mm3 can lead to splashing and a pronounced temperature gradient in the melting zone, causing pore formation. These phenomena, combined with increased heat penetration into the confined powder regions of the channel, collectively increase the channel surface roughness to 69.61 µm.
Comment -2: The author mentions that the surface roughness was tested 3 times, so an error bar should be included in Figure 2.
Response -2: Error bars have been added in Figures 2, 6, 8, 10, 12. For example, Figure 2 is updated as follows:
Figure 2. Building orientation effect on surface roughness of cuboid open surfaces and channel interior roof and floor.
Comment -3: The 3D arrangement of Figure 4c is difficult for the reader to discern, and the horizontal coordinate is obscured from view.
Response -3: The coordinates of Figure 4(c) have been rotated and adjusted for a clear view
Figure 4. Principal Component Statistical Analysis: (a) loading plot; (b) 2-D score plot of the first two principal components containing a cumulative 87.4% variance; (c) 3-D score plot of the first three principal components containing a cumulative 95% variance.
Comment -4: The horizontal coordinates of Figure 8,9,10, and 12 are not shown.
Response -4: Horizontal coordinates of Figures 6, 8, 10 and 12 have made clear and visible. For example, Figure 6 is updated as follows:
Figure 6. Effects of layer thickness on the channel surface roughness and density of 1mm fluidic channels at different scan speeds: (a) channel surface roughness variation at 60W laser power; (b) density variation at 60W laser power; (c) channel surface roughness variation at 120W laser power; (d) density variation at 120W laser power.
Reviewer 2 Report
Comments and Suggestions for Authors
The main topic addressed by the research is investigation of the influence of process parameters on the internal roughness and its correlation with the density of horizontally SLM printed TC4 titanium alloy fine flow fluid channels with a diameter of 1 millimeter. The most influencing factors, such as build orientation, layer thickness, scan speed, laser power, and hatch space were investigated.
The topic is relevant in the field of additive manufacturing. It provide valuable insights to deliver low surface roughness and higher density and optimize process parameters for SLMed TC4 alloy structural components with small circular channels.
A few studies have been conducted to fabricated millimeter sized interior channels using this AM technology and tried to control the channel shape deformation. In the reviewed article, the authors conducted an extensive research campaign to study (a) the effects of building orientation on the confined surface geometries, (b) the effects of process parameters on surface roughness of the horizontal interior flow channels and (c) effect of layer thickness, scan speed, laser power and hatch space on surface roughness and its co-relationship to the density. The results are well discussed and interpreted. The main interpretations were confirmed by literature sources.
Experimental research methods are well described with sufficient detail to allow others to replicate and build on published results.
The conclusions are consistent with the evidence and arguments presented.
The references cited in the article are consistent with the subject of the article. They present the current state of knowledge, most of them have been published in recent years.
Tables and Figures have appropriate captions. They were made in a clear way. I have no critical comments regarding the graphic side of the manuscript.
In general, the manuscript is well scientifically written with adequate English communication. In my opinion, the article deserves publication.
Detailed comments:
In the Introduction section, the authors provided a well-written background pointing out the problems of ensuring the appropriate quality of AM fabricated parts. Many works on AM have been reviewed. However, it is not clear whether the indicated problems apply to specific materials or to all materials treated with AM technique.
line 186. It is suggested to define all acronyms the first time you use them. The article should be made readable for non-specialized readers.
section 2.3. Why did the authors study average roughness Ra (linear parameter)? Surface scanning of the spatial parameter Sa could better reflect surface roughness.
It is suggested that the Ra parameter be consistently identified as ‘average roughness’ or by a standardized name for this parameter throughout the manuscript.
Does Figure 4 contain ellipses or ellipsoids?
The results of Random Forest Machine Learning modeling are presented. However, the modeling methods used were not described.
It is unknown why 'Layer Thickness', 'Scan Speed', 'Laser Power', etc., were capitalized in the text.
The error bars should be marked in Figures 6, 8 and 10.
Figure 9 shows the results of finite element modeling. However, no modeling details are presented: element type, mesh sensitivity analysis, boundary conditions, thermo-mechanical properties used in FE-based model, etc.
Typo in title of section no. 3.6.
Author Response
Title: Surface Roughness of Interior Fine Flow Channels in Selective Laser Melted Ti-6Al-4V Alloy Components
Shamoon Al Islam1,2,3, Liang Hao1,2,3,*, Zunaira Javaid1,2,3, Wei Xiong1,4, Yan Li1,2,3, Yasir Jamil5, Qiaoyu Chen1,2,3 and Guangchao Han1,2,4
Response
We deeply appreciate the time and effort you have invested in reviewing our manuscript. Your detailed and insightful feedback has significantly contributed to the enhancement of our work. Following your valuable suggestions, we have made the recommended changes. We hope that the revisions now accurately reflect the improvement.
Comment -1: In the introduction section, the authors provided a well-written background pointing out the problems of ensuring the appropriate quality of AM fabricated parts. Many works on AM have been reviewed. However, it is not clear whether the indicated problems apply to specific materials or to all materials treated with AM technique.
Response -1: Thank you for your insightful feedback regarding the clarity of the material-specific challenges addressed in our study. In response to your comment, we have revised the introduction section to clarify that the discussed issues primarily pertain to Ti-6Al-4V (TC4) alloy, our subject material. We have emphasized that while the problems such as surface roughness and shape deformation are common to various materials processed through additive manufacturing (AM), our research specifically investigates these challenges in the context of TC4 alloy. This focus is due to TC4's critical application in space and high-performance sectors where precision and material properties are paramount. We acknowledge that other materials also encounter similar challenges in AM; however, our literature review has revealed that the extent of research specifically addressing TC4 alloy's internal surface texture, particularly in fine flow channels, is not as comprehensive. Hence, our study aims to fill this gap by focusing on optimizing the surface roughness and roundness in interior fine flow channels of SLM-fabricated TC4 alloy components. The modifications made to the introduction are intended to make this focus clear and provide a solid foundation for the necessity and relevance of our work.
Comment -2: line 186. lt is suggested to define all acronyms the first time you use them. The article should be made readable for non-specialized readers.
Response -2: Acronym of Relative Humidity (RH) is defined in section 2.2, line 188.
Comment -3: section 2.3. Why did the authors study average roughness Ra (linear parameter)? Surface scanning of the spatial parameter Sa could better reflect surface roughness.
Response -3: Ra was chosen for its widespread acceptance and application across various industrial and research contexts, ensuring our findings are directly comparable and highly relevant to existing literature. We acknowledge the comprehensive nature of Sa for detailed surface characterization; however, for the scope and comparative nature of our study, Ra provided a reliable and standardized measure of surface roughness that aligns with the objectives and constraints of our research. This choice also facilitates quality control processes where Ra remains a predominant parameter. Anyhow, we appreciate your expert suggestion and intend to incorporate Sa in future studies for more nuanced surface analysis. To validate the pertinence of Ra in our work, we have cited a few relevant studies that also measure surface roughness using Ra, reinforcing its continued significance in the field;
1- Effect of Surface Modifications on Surface Roughness of Ti6Al4V Alloy Manufactured by 3D Printing, Casting, and Wrought. DOI: https://doi.org/10.3390/ma16113989
2- Innovative Post-Processing for Complex Geometries and Inner Parts of 3D-Printed AlSi10Mg Devices. DOI: https://doi.org/10.3390/ma16217040
3- Thermal and hydraulic performance of Al alloy-based 3D printed triangular microchannel heatsink governed by rough walls with graphene and alumina nanofluids as working liquid. DOI: 10.1088/1361-6439/ad2304
Comment -4: It is suggested that the Ra parameter be consistently identified as 'average roughness' or by a standardized name for this parameter throughout the manuscript
Response -4: We have standardized the nomenclature for the roughness parameter (Ra) throughout the manuscript. “Roof roughness” and “Floor roughness” now consistently refer to the average roughness at the channel's roof and floor, respectively, as defined in Section 2.3. Additionally, “Channel surface roughness” is used to denote the average of the roof and floor roughness, with this standardization also detailed in Section 2.3. To ensure consistency and avoid confusion, we have made corresponding corrections in the discussion sections, aligning with the terminology. These changes can be seen in Sections 3.1 to 3.6 and lines 219, 222, 225, 266, 270, 320, 321, 336-358, 367-375, and 392-410 through 900-917."
Comment -5: Does Figure 4 contain ellipses or ellipsoids?
Response -5: Figure 4(c) contains ellipsoids, not ellipses. We have made the relevant corrections in Section 3.4, specifically in lines 408 and 409. Additionally, the coordinates of Figure 4(c) have been adjusted to clearly display the ellipsoids.
Figure 4. Principal Component Statistical Analysis: (a) loading plot; (b) 2-D score plot of the first two principal components containing a cumulative 87.4% variance; (c) 3-D score plot of the first three principal components containing a cumulative 95% variance.
Comment -6: The results of Random Forest Machine Learning modeling are presented. However, the modeling methods used were not described.
Response -6: In response to your valuable comment, we have added a comprehensive explanation of our Random Forest Machine Learning (RF-ML) regression model to Section 2.3, specifically from lines 265 to 303. This addition details our supervised learning approach, data preprocessing steps, model configuration, training process, and performance evaluation metrics. We have outlined how the model was trained on labeled data with the target variable being the measured channel surface roughness for each set of process parameters (laser power, scan speed, and layer thickness). Our description includes the meticulous development of the RF-ML model, emphasizing data normalization, the configuration of the model's trees and depth, the employment of bootstrapping, and the use of cross-validation to ensure model generalizability and accuracy. We believe that this detailed methodological explanation addresses the previous lack of clarity and provides a solid understanding of how the RF-ML model was employed to examine parameter interactions and optimize surface roughness in SLM-fabricated Ti-6Al-4V alloy components.
Comment -7: It is unknown why 'Layer Thickness’, 'Scan Speed', 'Laser Power', etc., were capitalized in the text.
Response -7: The capitalization of “Layer Thickness”, “Scan Speed”, “Laser Power”, etc., was unintentional and has been corrected to ensure consistency throughout the text. These revisions can be found in Section 3.2, specifically at lines 429, 432, and 437. We appreciate your keen eye for detail, which has helped improve the manuscript.
Comment -8: The error bars should be marked in Figures 6, 8 and 10.
Response -8: Error bars have been added in Figures 2, 6, 8, 10, 12. For example, Figure 6 is updated as follows:
Figure 6. Effects of layer thickness on the channel surface roughness and density of 1mm fluidic channels at different scan speeds: (a) channel surface roughness variation at 60W laser power; (b) density variation at 60W laser power; (c) channel surface roughness variation at 120W laser power; (d) density variation at 120W laser power.
Comment -9: Figure 9 shows the results of finite element modeling. However, no modeling details are presented: element type, mesh sensitivity analysis, boundary conditions, thermo-mechanical properties used in FE-based model. etc.
Response -9: Thank you for your valuable feedback regarding the presentation of finite element modeling details in Figure 9. We have expanded the methodology (section 2.3, lines 229 to 259) to include comprehensive details about the finite element analysis (FEA) process employed in our study. This includes a thorough description of the thermal manufacturing simulations conducted using Simufact Additive by Hexagon, the determination of TC4 material properties via JMatPro V7.0 software, and a detailed account of our comprehensive mesh sensitivity analysis. We have clarified the selection of element types, boundary conditions, and thermo-mechanical properties critical to the simulation's fidelity for SLM processes. Additionally, we have described the procedural steps taken from geometry importation through to post-processing, ensuring clarity on the simulation parameters such as hatch space, laser power, scan speed, layer thickness, and scan strategy. We believe these additions comprehensively address the initial concern and provide the necessary modeling details to support the results presented in Figure 9.
Comment -10: Typo in title of section no. 3.6
Response -10: Thank you for pointing out the typo in the title of Section 3.6. We have corrected this error, as can be seen in line 826. We appreciate your attention to detail, which has helped improve the clarity and accuracy of our manuscript.